# Higher Waist Hip Ratio Genetic Risk Score Is Associated with Reduced Weight Loss in Patients with Severe Obesity Completing a Meal Replacement Programme

**DOI:** 10.3390/jpm12111881

**Published:** 2022-11-09

**Authors:** Dale Handley, Mohammed Faraz Rafey, Sumaya Almansoori, John F. Brazil, Aisling McCarthy, Hasnat A. Amin, Martin O’Donnell, Alexandra I. Blakemore, Francis M. Finucane

**Affiliations:** 1College of Health, Medical and Life Sciences, Brunel University London, London UB8 3PH, UK; 2Bariatric Medicine Service, Centre for Diabetes, Endocrinology and Metabolism, Galway University Hospitals, H91 YR71 Galway, Ireland; 3HRB Clinical Research Facility, University of Galway, H91 CF50 Galway, Ireland; 4Department of Medicine, University of Galway, H91 CF50 Galway, Ireland; 5Faculty of Medicine, Imperial College London, London SW7 2BX, UK; 6International Centre for Forensic Science, General Department of Forensic Science and Criminology, Dubai Police, Dubai 00000, United Arab Emirates

**Keywords:** bariatric, body mass index, genetic risk score, meal replacement, milk, severe obesity, single nucleotide polymorphism, waist–hip ratio

## Abstract

**Background:** A better understanding of the influence of genetic factors on the response to lifestyle interventions in people with obesity may allow the development of more personalised, effective and efficient therapeutic strategies. We sought to determine the influence of six obesity-related genetic risk scores on the magnitude of weight lost by patients with severe obesity who completed a dietary intervention. **Methods:** In this single-centre prospective cohort study, participants with severe and complicated obesity who completed a 24-week, milk-based meal replacement programme were genotyped to detect the frequency of common risk alleles for obesity and type 2 diabetes-related traits. Genetic risk scores (GRS) for six of these traits were derived. Participants with a potentially deleterious monogenic gene variant were excluded from the analysis. **Results:** In 93 patients completing the programme who were not carrying a known obesity-related gene mutation, 35.5% had diabetes, 53.8% were female, mean age was 51.4 ± 11 years, mean body mass index was 51.5 ± 8.7 and mean total weight loss percent at 24 weeks was 16 ± 6.3%. The waist–hip ratio (WHR) GRS was inversely associated with percentage total weight loss at 24 weeks (adjusted β for one standard deviation increase in WHR GRS −11.6 [−23.0, −0.3], *p* = 0.045), and patients in the lowest tertile of WHR GRS lost more weight. **Conclusions:** Patients with severe and complicated obesity with a genetic predisposition to central fat accumulation had less weight loss in a 24-week milk-based meal replacement programme, but there was no evidence for influence from the five other obesity-related genetic risk scores on the response to dietary restriction.

## 1. Background

For people with severe obesity (conventionally defined as a body mass index (BMI) ≥40 kg m^−2^ or ≥35 kg m^−2^ with co-morbidities such as type 2 diabetes), relatively intensive weight loss interventions are often indicated [1]. While bariatric surgery is a very effective treatment for obesity and related disorders [2], it is not suitable for all patients. Similarly, there is heterogeneity in the need for and response to obesity drug therapy [3]. Lifestyle modification is always the cornerstone of the therapeutic approach to the patient with obesity. Several studies have confirmed the benefits of structured lifestyle interventions in different patient groups, including those with non-diabetic hyperglycaemia [4], cardiovascular disease [5] and type 2 diabetes [6]. However, meaningful, sustained reductions in weight over time are difficult to achieve with lifestyle approaches alone [7]. Some suggest that a meaningful improvement in health requires a weight loss of 10% [8], although we have recently described improvements in fitness and reductions in blood pressure, lipid profiles and HbA1c after more modest weight loss in a prospective cohort study of patients with severe obesity attending our service [9].

Whilst environmental factors such as sedentary lifestyle and poor diet are implicated in the pathogenesis of obesity and type 2 diabetes, complex genetic factors also modify disease expression [10]. Significant differences have been observed in diabetes prevalence rates between distinct ethnic groups and these are not accounted for by environmental factors alone [11]. Additionally, there is a high rate of disease concordance in monozygotic twins [12]. A study comparing differences in concordance between monozygotic and dizygotic twins reported a heritability estimate of 77% for type 2 diabetes and 65% for body mass index (BMI) [13]. Heritability estimates for obesity vary according to BMI, being lower in the overweight range and higher with normal weight or obesity [14]. Further evidence of the importance of heritability in the development of the metabolic consequences of obesity comes from studies in individuals who have a family history of diabetes. Normal glucose tolerant offspring who have a parent with diabetes have reduced skeletal muscle oxygen uptake in spite of similar physical activity levels to those without a family history [15]. These individuals have also been found to have impaired mitochondrial function [16], insulin sensitivity [17] and beta-cell function [18].

Genome-wide association studies (GWAS) examining several hundred thousand single nucleotide polymorphisms (SNPs) in large populations have identified common genetic variants associated with obesity. The first of these was the fat mass and obesity associated (FTO) gene, for which the risk allele predisposes to diabetes through an effect on BMI [19]. Adults who are homozygous for the risk allele weigh approximately 3 kg more and have a 1.67-fold increased risk of obesity compared to those with no risk allele [20]. Further GWAS studies have revealed more than 100 different common genetic variants or regions associated with BMI [21], with almost all of these (including many components of the melanocortin pathway) acting in the central nervous system and influencing food intake and dietary behaviour [22]. How polymorphisms modify the relationship between anthropometric and metabolic traits and lifestyle factors such as diet and physical activity has not been fully established. Several researchers have described the influence of genetic polymorphisms on the response to lifestyle modification [23,24]. In the Finnish Diabetes Prevention Study, the Pro12Ala polymorphism conferred a two-fold increased risk in the overall cohort of progression to diabetes [25]. In the US Diabetes Prevention Program, TCF7L2 polymorphisms were associated with an increased risk of progression to diabetes [26], such that the beneficial effects of lifestyle were abolished in the TCF7L2 risk allele group.

Several researchers have described the use of low energy liquid diets as components of intensive lifestyle modification programmes for the treatment of obesity. Typical initial weight loss is approximately 10 kg [27,28,29], but often weight regain limits the longer-term efficacy of these interventions [7] and retention rates are low [30]. At the regional endocrinology clinic for patients with severe obesity in Galway University Hospitals, we have developed a 24-week, outpatient, milk-based meal replacement programme for patients with severe obesity. There is an initial weight loss phase, followed by weight stabilisation and weight maintenance phases, each lasting eight weeks, with an average total body weight loss in programme completers of 15.9 ± 6.0% [31]. This cohort of patients offered a unique opportunity to examine the influence of common SNPs identified using genome-wide association studies for obesity-related traits on the response to an intensive milk-based meal replacement programme in adults with severe obesity.

## 2. Methods

### 2.1. Study Design

The “GERONIMO (Genetic Effects on the Response to an Outpatient Intensive Nutritional Intervention in Medically Complicated Obesity) Study” was a single-centre, retrospective cohort study of adults with severe obesity attending our hospital-based endocrinology service, who completed our milk-based meal replacement programme.

### 2.2. Setting

All metabolic and anthropometric baseline and follow-up measures for the study were conducted at the Centre for Diabetes, Endocrinology and Metabolism at Galway University Hospitals. Each visit during the milk programme also took place there. The acquisition of blood for genotyping took place at the Health Research Board Clinical Research Facility (HRB CRF) in GUH. Approval to conduct the study was provided by the Galway University Hospitals’ Central Research Ethics Committee (reference CA-1802) in November 2018. The study was conducted according to STROBE (Strengthening The Reporting of OBservational Studies in Epidemiology) guidelines [32].

### 2.3. Study Population

Participants were recruited from the already established cohort of milk programme completers as outlined above, who were attending for follow-up. An invitation letter and information sheet were prepared, detailing the requirements of study participation and the procedures involved. For those who agreed to participate, an appointment was made to attend the HRB CRFG after an overnight fast. All procedures were carried out in accordance with the principles of good clinical practice. Fully informed written consent was obtained from each participant prior to testing.

### 2.4. Inclusion and Exclusion Criteria

Male and female patients aged 18 years or older, attending the endocrinology service for management of severe and complicated obesity and who completed the milk-based meal replacement programme were eligible for inclusion. Severe obesity was defined as a BMI ≥40 kg m^−2^ (or ≥35 kg m^−2^ with co-morbidities such as type 2 diabetes or obstructive sleep apnoea syndrome). Female patients of childbearing potential who were pregnant, breast-feeding or intended to become pregnant or were not using adequate contraceptive methods were excluded from the original milk diet intervention. Those with a recent myocardial infarction (within six months), untreated arrhythmia, untreated left ventricular failure, recent cholelithiasis (within one year), hepatic or renal dysfunction, type 1 diabetes, major psychiatric disorders, eating disorders, cancer, previous bariatric surgery, a BMI <35 kg m^−2^ or those deemed unlikely to attend for the full programme (e.g., frequent clinic non-attendance) were also excluded from the milk diet intervention.

### 2.5. Milk Diet Intervention

The milk-based low energy liquid diet (LELD) consisted of three continuous eight-week phases, each with fortnightly visits to the endocrinology clinic. During the first (weight loss) phase from weeks one to eight inclusive, an exclusively milk-based liquid diet was prescribed, with approximately 2.5 L per day of semi-skimmed milk, divided over seven equal portions throughout the day, with additional sodium, vitamin, mineral and fibre supplementation. This was equivalent to approximately 1200 kcal, 130 g of carbohydrates and 40 g of fat per participant per day. The specific caloric content and amount of milk was calculated according to each participant’s weight at baseline and their estimated daily protein requirements (calculated using the formula 0.17 g N_2_/kg × 6.25). For patients with a BMI < 50 kg m^−2^, we replaced 75% of their daily protein requirements and for those with a BMI ≥ 50 kg m^−2^, 65% of their total daily protein requirements were replaced, equivalent to approximately 130 g of protein per day, as we have described previously [31]. During the second phase (weight stabilization) from weeks nine to sixteen inclusive, there was a gradual re-introduction of low-calorie meals from a set menu over eight weeks, under the supervision of the dietitian. Finally, in the third phase (weight maintenance) from weeks 17 to 24 inclusive, the milk component of the diet was stopped completely and a fully solid isocaloric diet was established, with individualized meal plans, under dietetic supervision.

### 2.6. Anthropometric Measurements

Weight was measured on a Tanita^®^ scale and height with a Seca^®^ wall-mounted stadiometer.

### 2.7. Blood Samples

Bloods taken during the milk diet intervention were drawn after an overnight fast. All blood samples were processed locally in the Galway University Hospitals’ Department of Clinical Biochemistry (certified to ISO 15189 2007 accreditation standard). HbA1c was measured with high pressure liquid chromatography (HPLC) (Menarini^®^ (Florence, Italy) HA8160 auto-analyzer). Total cholesterol was measured using the CHOP-PAP method. HDL-cholesterol and triglycerides were measured using the enzymatic and the GPO-PAP methods, respectively (COBAS^®^ 8000 modular analyser (Roche, West Sussex, U.K.)). LDL-Cholesterol was derived with the Friedewald equation.

### 2.8. Genotyping

After fully informed written consent for genotyping, an ethylenediamine tetra-acetic acid (EDTA) sample was drawn and sent to the genetics laboratory by courier. Genotyping was performed using a custom Axiom genotyping array prepared by ThermoFisher. Briefly, the genotyping array was designed to detect 114,782 variants, 97.2% of which were rare variants in candidate genes for monogenic obesity and diabetes. The remainder were specific common SNPs identified from obesity-related GWAS studies, in order to generate genetic risk scores as used in this study. Sample genotyping was performed for 101 samples according to Thermo Fisher guidelines by Oxford genomics and processed by the GeneTitan Multi-channel instrument (Thermo Fisher Scientific, Waltham, Massachusetts, USA). Probe clustering was performed using the AxiomGT1 algorithm in the Axiom analysis suite V5.1.1. Rare heterozygosity adjustment was used to adjust for multi-probe mismatches, which substantially improves correct rare variant calls for very rare variants [33]. Initial genotyping filtering was conducted according to the Axiom array guidelines, with the per-sample call rate set to 93% and the dish quality control (DQC) threshold set to 82%, in order to detect poor quality genotyping due to the presence of a high background signal resulting from erroneous sample preparation or DNA contamination. Data from samples that did not pass the quality control (QC) metrics provided above (*n* = 0), were duplicates (*n* = 0), that showed a high degree of heterozygosity or relatedness (*n* = 0), or which displayed discordant genetic and self-reported sex (*n* = 0) were removed from further analysis. Individuals were deemed to have a potentially deleterious monogenic obesity variant if they carried either two rare (minor allele frequency <1%) autosomal recessive variants or one autosomal dominant variant with a combined annotation dependent depletion (CADD) score above 15 in known or suspected monogenic obesity genes [34]. Such individuals were removed from the analyses reported here. Common SNPs, as identified from available summary statistics in European individuals of previously conducted GWAS, for body mass index [35], body fat percentage [36], favourable adiposity [37], waist–hip ratio (WHR) (adjusted for BMI) [38], adiponectin [39] and type 2 diabetes [40] were used to construct the respective weighted genetic risk scores for each phenotype using Plink v2.0 [41]. Any variants which had a variant-wise missingness above 5% were removed prior to deriving the genetic risk scores (*n* = 0 across all scores).

### 2.9. Statistical Analysis Plan

A comparison of patient characteristics at baseline according to diabetes status was performed using the unpaired *t*-test for normally distributed continuous variables and the Wilcoxon Rank Sum Test (i.e., the Mann Whitney U Test) for non-normally distributed continuous variables. Differences in proportions were assessed using the Chi-squared test. Differences in the magnitude of weight loss during the milk diet between genetic risk score tertiles were assessed using one way analysis of variance (ANOVA), including pairwise ANOVA with Bonferroni adjustment. Associations between genetic risk scores (treated as a continuous independent or exposure variable) and weight change (as the continuous dependent or outcome variable) were measured using linear regression, including adjustment for age, sex, ethnicity, diabetes status and diabetes medication usage. Stata SE^®^ Version 17 (College Station, Texas, USA) was used for all statistical analyses.

## 3. Results

Between January 2013 and October 2018, we reviewed 1867 newly referred patients with severe obesity in our endocrinology clinic, as outlined in Figure 1. Of these, 260 patients (13.9%) started the milk-based meal replacement programme and of these, 139 (53.5%) completed all 24 weeks of the intervention, while 121 (46.5%) discontinued (“dropped out of”) the intervention and were not included in these analyses. Of the 139 milk diet completers we invited to this study, 105 (75.5%) agreed to participate and provided written informed consent. In four of these patients, the blood samples obtained were of insufficient volume or quality, or were missing, which precluded genetic analysis. In the remaining 101 patients, genotyping revealed a gene mutation implicated in obesity pathogenesis in eight patients, and these patients were excluded from further analysis here. The baseline characteristics of the remaining 93 patients are shown in Table 1, according to their diabetes status. The 35% of patients with diabetes had similar age, sex and blood pressure to those without diabetes, with lower BMI and excess body weight, consistent with a lower weight threshold for starting this intervention compared to patients without diabetes. Variations in lipid profiles were likely due to a higher prevalence of statin use in patients with diabetes (though we do not have information on statin use in these patients to hand). Baseline differences in HbA1c and diabetes medication use were as anticipated. Patients were predominantly of White Irish self-reported ethnicity, with Polish White (two), Irish Traveller (two), German Jewish White (one), German White (one), Asian Pakistani (one) and Hungarian White (one) also reported. Patients with diabetes had a slightly higher type 2 diabetes genetic risk score, but other scores were similar in patients with and without diabetes. In the cohort overall, the excess body weight percentage at the start of the milk diet was 106.1 ± 34.9% (range 36.9–187.1%). This reduced to an excess body weight percentage of 73 ± 31.9% (range 10.5–142.7%) after 24 weeks (*p* < 0.0001). The absolute change in excess body weight percentage in the overall cohort was 33.1 ± 13.8% (range 9.1–68.3%).

Of the six genetic risk scores we tested, two (BMI GRS and Adiponectin GRS) were associated with baseline weight and BMI in adjusted regression models, but the other four scores were not, as shown in Table 2. Next, we categorised the genetic risk scores according to their tertile. When responses to the milk-based meal replacement programme after 24 weeks according to the tertile of the relevant genetic risk score were compared using one-way analysis of variance, the magnitude of the reduction in total and excess body weight percentage was greater in patients who were within the lowest tertile of WHR GRS, compared to higher tertiles, as shown in Figure 2a–e. However, there were no statistically significant differences in the magnitude of percentage excess weight loss according to tertile for any of the other five genetic risk scores, nor was the WHR GRS associated with baseline weight or BMI. In order to account for the potential confounding effects of age, sex, diabetes status, ethnicity and diabetes medication usage, we treated the six genetic risk scores as (separate) continuous exposure (or independent) variables and indices of weight loss as outcome (or dependent) variables, in unadjusted and adjusted regression models as shown in Table 2. Once again, the only genetic risk score that was associated with the magnitude of weight loss was the WHR GRS. For every 0.1 unit increase in the WHR GRS, the BMI at follow up (adjusted for baseline BMI and the above confounders) was 0.58 kgm^−2^ higher, weight was 1.65 kg higher and the total reduction in body weight was 1.16% lower. The absolute change in percentage excess body weight tended towards being 2.41% lower for every 0.1 unit increase in WHR GRS, but this association was not statistically significant (*p* = 0.06), as shown. Results were similar in adjusted and unadjusted analyses.

For Figure 2a–e, the *p*-values for between-tertile comparisons were derived from pairwise ANOVA, with Bonferroni correction.

## 4. Discussion

We have shown that in a cohort of adults with severe obesity completing a 24−week milk-based meal replacement programme, the magnitude of weight loss was greater in those participants in the lowest tertile of WHR GRS and that the inverse association between WHR GRS and weight loss persisted after adjusting for age, sex and diabetes medication usage. This suggests that those people with severe obesity who have a genetic tendency to store fat centrally respond less well to intensive dietary restriction. To our knowledge, this observation has not been described previously in patients with severe obesity undergoing a meal replacement programme. Populations with severe obesity are known to carry a higher genetic risk burden for body fatness than the general population and are likely to be enriched with risk alleles for obesity [42], and previous research has examined how SNPs in candidate genes influence their response to other intensive weight loss interventions, such as bariatric surgery. For example, variations in SNPs at FKB51 [43], MC4R [44] and FTO [45] influence the magnitude and timing of maximal weight loss after bariatric surgery, though other studies have suggested that genetic influences on post-operative weight loss by genotype can be difficult to elucidate and are likely to be subtle [46].

Our results are consistent with findings in other studies of the influence of polygenic risk scores derived from multiple SNPs on the response to lifestyle interventions. For example, a study of US hospital workers undergoing a “healthy eating advice” intervention found that those with a higher BMI GRS gained more weight and made less healthy choices in the hospital canteen over two years than those with lower scores [47]. More recently, a consortium analysed variations in short- and medium-term weight loss outcomes from seven separate lifestyle modification trials, noting that participants with a higher waist circumference-related GRS had smaller reductions in central adiposity [48]. This is consistent with observations in studies of genetic influences on the response to bariatric surgery, where patients with specific risk alleles for obesity have been shown to have lower weight loss post-operatively [43,44,45]. While some have advocated the adoption of obesity genetic risk scores to predict which patients will respond to bariatric surgery [49], they do not perform as well as clinical prediction models (based on factors such as age, surgery type and diabetes status) and offer only marginal enhancements to receiver operating characteristic curves when combined with clinical variables [50].

Our study has several strengths. The mean effect size of the intervention was relatively large and achieved over a precisely defined timeframe. Also, heterogeneity in the intervention “exposure” was relatively modest. Furthermore, the cohort consisted of predominantly White Irish patients, so while the generalisability of the results to other patient groups is limited, we have kept the important but mechanistically less relevant confounding effect of ethnic heterogeneity to a minimum, insofar as possible. Nonetheless, ethnicity has been an important factor in other studies: The consortium of lifestyle modification trials mentioned above noted that the effects of the waist circumference GRS on the response to lifestyle interventions was only apparent in White participants [48]. They also noted that the overall effect of the GRS was not clinically significant. These findings provide preliminary but convincing evidence of an association between the WHR GRS and weight loss outcomes in patients with severe obesity undergoing dietary restriction. A limitation of our study is the absence of clinical information on important confounding factors such as thyroid dysfunction, steroid use or immobility. However, we think that these would introduce random error and imprecision to our results rather than providing a false indication of an association between genetic risk scores and weight loss, where none existed. Future prospective studies can address these limitations. These findings require further exploration in other ethnic groups and in those undergoing different interventions such as bariatric surgery or treatment with medications. Determining the impact of GRS scores on obesity-related metabolic and vascular outcomes in larger, longer-term studies will require more detailed phenotypic assessment at baseline and follow-up, but seems warranted. Ultimately this might help to broaden and refine the range of therapeutic options that are available for adults affected by severe obesity.

## Figures and Tables

**Figure 1 jpm-12-01881-f001:**
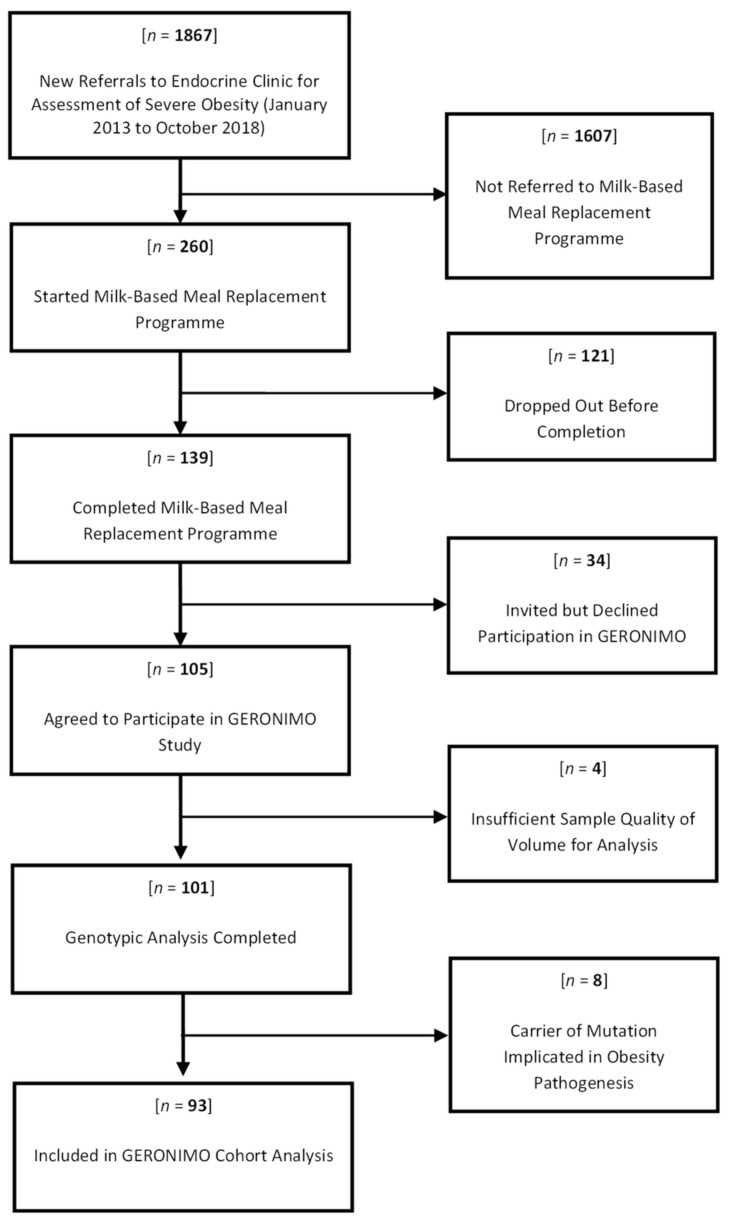
Flow Diagram of Recruitment of Participants to GERONIMO Study.

**Figure 2 jpm-12-01881-f002:**
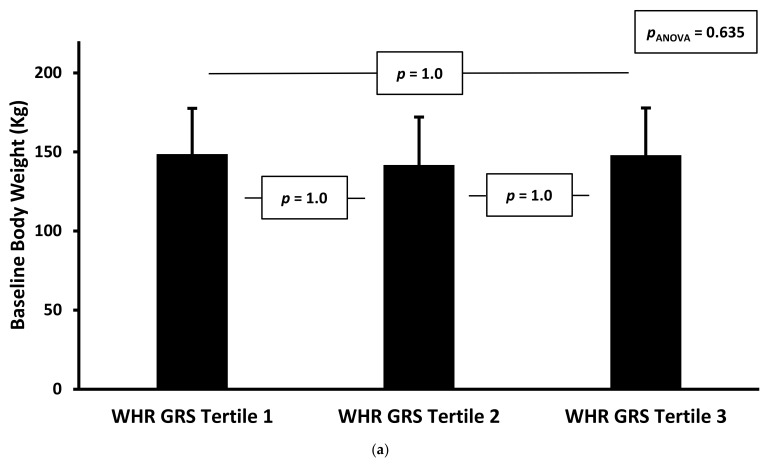
(**a**) Baseline Body Weight, By Tertile of Waist–Hip Ratio GRS. (**b**) Baseline Body Mass Index, By Tertile of Waist–Hip Ratio GRS. (**c**) Total Body Weight Loss Percentage at 24 Weeks, By Tertile of Waist–Hip Ratio GRS. (**d**) Excess Body Weight Loss Percentage at 24 Weeks, By Tertile of Waist–Hip Ratio GRS. (**e**) Total Body Weight Loss in Kilograms at 24 Weeks, By Tertile of Waist–Hip Ratio GRS.

**Table 1 jpm-12-01881-t001:** Baseline Characteristics of GERONIMO Study Participants, According to Diabetes Status.

	Type 2 Diabetes		No Diabetes		*p*-Value
*n*	33		60		
Age (years)	52.7	±8.9	50.6	±12	0.39
Sex (female)	15	(45.5%)	35	(58.3%)	0.23
Ethnicity (White Irish)	30	(90.9%)	54	(90%)	0.62
Height (m)	1.69	±0.1	1.68	±0.1	0.67
Weight (kg) *	131	(117.0, 159.4)	145.3	(130, 167.1)	0.056
Ideal Body Weight (kg) *	70.6	(65.6, 76.6)	68.9	(64.4, 76.6)	0.67
Excess Body Weight (kg) *	56.8	(44.6, 91.3)	76.5	(64.1, 94.3)	0.031
Excess Body Weight (%)	95.9	±40.6	111.6	±30.3	0.05
Body Mass Index (kgm^−2^)	49	±10.1	52.9	±7.6	0.037
Total Cholesterol (mmol/l)	4.3	±1.2	4.9	±0.8	0.0042
LDL-Cholesterol (mmol/l)	2.3	±0.9	2.8	±0.8	0.0033
HDL-Cholesterol (mmol/l) *	1	(0.9, 1.2)	1.2	(1.1, 1.5)	0.0045
Triglycerides (mmol/l) *	1.9	(1.5, 2.5)	1.6	(1.3, 1.9)	0.047
Triglyceride: HDL Ratio *	4.6	(3.3, 5.6)	3.1	(2.1, 3.8)	0.0019
HbA1c (mmol/mol)	67	(55, 73)	39	(36, 41)	<0.001
GRS—Body Mass Index	2.06	±0.16	2.07	±0.14	0.62
GRS—Body Fat Percent	0.35	±0.07	0.35	±0.07	0.95
GRS—Favourable Adiposity	0.17	±0.03	0.17	±0.03	0.64
GRS—Waist–Hip Ratio	1.14	±0.10	1.14	±0.13	0.83
GRS—Type 2 Diabetes	10.25	±0.40	10.01	±0.49	0.022
GRS—Adiponectin	0.28	±0.08	0.30	±0.06	0.074
On metformin	30	(90.9%)	7	(11.7%)	<0.001
On insulin	9	(27.3%)	0	(0%)	<0.001
On sulphonylurea	13	(39.4%)	0	(0%)	<0.001
On DPPIV-inhibitor	6	(18.2%)	0	(0%)	0.001
On SGLT2-inhibitor	6	(18.2%)	0	(0%)	0.001
On GLP1-receptor agonist	13	(39.4%)	3	(5%)	<0.001
On PPARγ-receptor agonist	1	(3%)	0	(0%)	0.18

DPPIV: Dipeptidyl Peptidase IV, GLP1: Glucagon-Like Peptide, GRS: Genetic Risk Score, PPARγ: Peroxisome Proliferator-Activated-Receptor Gamma, SGLT2: Sodium-Glucose-Like Transporter-2. Normally distributed data are presented as means ± standard deviation. * Non-normally distributed data are presented as median (interquartile range). Numbers of participants in each group are presented as *n* (percentage). Comparisons between normally distributed variables were made using the unpaired *t*-test and between non-normally distributed variables using the Wilcoxon Rank Sum Test (i.e., the Mann Whitney U test). Differences in proportions were assessed using the Chi-squared test.

**Table 2 jpm-12-01881-t002:** Associations Between Obesity-Related Genetic Risk Scores and Changes after 24 Weeks in Anthropometric and Metabolic Variables in Patients with Severe Obesity Completing the Milk-Based Meal Replacement Programme.

Variable	Model 1			Model 2			Model 3		
	β	[95% C.I.]	*p*	β	[95% C.I.]	*p*	β	[95% C.I.]	*p*
BMI GRS Score:									
Baseline Weight (kg)	30.0	[−10.9, 70.8]	0.148	27.3	[−6.2, 60.8]	0.108	36.5	[2.3, 70.7]	0.037
Baseline BMI (kg m^−2^)	10.6	[−1.6, 22.8]	0.087	10.5	[−1.3, 22.2]	0.081	13.0	[0.8, 25.3]	0.038
Follow-up Weight (kg)	−5.6	[−18.8, 7.6]	0.401	−6.2	[−19.5, 7.0]	0.353	−5.5	[−20.1, 9.0]	0.451
Follow-up BMI (kg m^−2^)	−2.4	[−7.0, 2.3]	0.312	−2.3	[−7.0, 2.3]	0.319	−2.0	[−7.1, 3.1]	0.441
Change in Total Body Weight (%)	4.5	[−4.5, 13.5]	0.328	4.3	[−4.7, 13.3]	0.345	3.1	[−6.7, 12.8]	0.532
Change in Excess Body Weight (%)	16.2	[−3.5, 36.0]	0.106	15.8	[−3.8, 35.4]	0.112	15.6	[−5.4, 36.6]	0.143
Body Fat Percentage GRS Score:									
Baseline Weight (kg)	60.8	[−20.2, 141.9]	0.139	42.9	[−24.6, 110.5]	0.210	37.9	[−29.5, 105.3]	0.266
Baseline BMI (kg m^−2^)	14.6	[−9.9, 39.2]	0.239	15.7	[−8.1, 39.5]	0.193	14.5	[−9.7, 38.6]	0.237
Follow-up Weight (kg)	−11.8	[−37.8, 14.2]	0.368	−11.3	[−37.5, 14.8]	0.392	−9.6	[−37.2, 17.9]	0.488
Follow-up BMI (kg m^−2^)	−4.4	[−13.5, 4.7]	0.336	−3.7	[−12.9, 5.5]	0.424	−3.0	[−12.6, 6.7]	0.545
Change in Total Body Weight (%)	7.2	[−10.6, 24.9]	0.426	5.6	[−12.3, 23.5]	0.537	3.8	[−15.1, 22.6]	0.693
Change in Excess Body Weight (%)	26.7	[−12.1, 65.6]	0.175	24.3	[−14.5, 63.1]	0.217	20.4	[−20.3, 61.1]	0.321
Favourable Adiposity GRS Score:									
Baseline Weight (kg)	−9.9	[−205.8, 186.1]	0.921	8.8	[−157.8, 175.5]	0.916	−21.0	[−191.1, 149.0]	0.806
Baseline BMI (kg m^−2^)	8.9	[−49.8, 67.5]	0.765	9.6	[−47.2, 66.4]	0.738	1.3	[−57.9, 60.6]	0.965
Follow-up Weight (kg)	2.7	[−60.4, 65.8]	0.931	0.8	[−62.2, 63.8]	0.979	6.1	[−61.9, 74.2]	0.858
Follow-up BMI (kg m^−2^)	0.0	[−22.1, 22.0]	0.998	−0.8	[−22.8, 21.2]	0.944	−0.3	[−24.2, 23.6]	0.980
Change in Total Body Weight (%)	−3.9	[−47.6, 39.7]	0.858	−2.5	[−46.0, 41.0]	0.909	−4.3	[−51.3, 42.7]	0.857
Change in Excess Body Weight (%)	5.8	[−89.7, 101.3]	0.904	9.2	[−85.3, 103.7]	0.847	2.0	[−99.3, 103.2]	0.969
Waist–Hip Ratio GRS Score:									
Baseline Weight (kg)	−3.2	[−56.3, 49.9]	0.905	7.3	[−37.3, 52.0]	0.745	10.8	[−32.4, 54.1]	0.619
Baseline BMI (kg m^−2^)	−3.0	[−19.2, 13.1]	0.709	−1.6	[−17.2, 13.9]	0.835	−1.2	[−16.6, 14.2]	0.877
Follow-up Weight (kg)	15.1	[−0.5, 30.7]	0.057	14.5	[−1.3, 30.2]	0.071	16.5	[0.0, 33.0]	0.050
Follow-up BMI (kg m^−2^)	5.5	[0.0, 10.9]	0.051	5.3	[−0.3, 10.8]	0.062	5.8	[0.0, 11.6]	0.050
Change in Total Body Weight (%)	−10.8	[−21.6, 0.0]	0.049	−10.4	[−21.3, 0.5]	0.060	−11.6	[−23.0, −0.3]	0.045
Change in Excess Body Weight (%)	−23.9	[−48.2, 0.4]	0.054	−22.1	[−46.3, 2.0]	0.072	−24.1	[−49.2, 1.0]	0.060
Adiponectin GRS Score:									
Baseline Weight (kg)	137.3	[49.7, 225.0]	0.002	100.7	[24.4, 177.0]	0.010	46.7	[−39.4, 132.7]	0.283
Baseline BMI (kg m^−2^)	36.0	[9.1, 62.9]	0.009	36.1	[9.6, 62.5]	0.008	22.3	[−8.1, 52.6]	0.149
Follow-up Weight (kg)	−14.0	[−44.7, 16.6]	0.366	−13.4	[−44.1, 17.3]	0.386	−10.0	[−44.6, 24.7]	0.569
Follow-up BMI (kg m^−2^)	−5.7	[−16.3, 4.8]	0.285	−4.3	[−15.0, 6.4]	0.429	−3.9	[−16.1, 8.4]	0.532
Change in Total Body Weight (%)	6.7	[−13.5, 26.9]	0.512	3.7	[−16.8, 24.2]	0.722	2.3	[−21.5, 26.1]	0.850
Change in Excess Body Weight (%)	43.8	[0.5, 87.2]	0.047	38.1	[−5.7, 81.9]	0.088	27.9	[−22.9, 78.7]	0.278
Type 2 Diabetes GRS Score:									
Baseline Weight (kg)	−4.5	[−17.2, 8.1]	0.479	−2.5	[−13.4, 8.4]	0.649	−1.3	[−12.6, 10.0]	0.818
Baseline BMI (kg m^−2^)	−0.2	[−4.1, 3.6]	0.901	−0.9	[−4.7, 2.9]	0.633	−0.6	[−4.6, 3.4]	0.765
Follow-up Weight (kg)	3.4	[−0.5, 7.3]	0.086	3.3	[−0.8, 7.3]	0.111	3.5	[−1.0, 7.9]	0.123
Follow-up BMI (kg m^−2^)	1.2	[−0.2, 2.6]	0.090	1.1	[−0.3, 2.5]	0.133	1.2	[−0.4, 2.7]	0.138
Change in Total Body Weight (%)	−2.2	[−4.9, 0.5]	0.109	−2.0	[−4.8, 0.8]	0.160	−2.3	[−5.4, 0.8]	0.140
Change in Excess Body Weight (%)	−4.9	[−10.9, 1.1]	0.109	−4.9	[−10.9, 1.2]	0.114	−5.1	[−11.7, 1.6]	0.134
Change in Excess Body Weight (%)	−10.4	[−47.3, 26.4]	0.575	−10.7	[−47.1, 25.7]	0.641	−11.4	[−51.2, 28.3]	0.567

β coefficients and [95% confidence intervals] are presented with the relevant genetic risk score as the exposure (or independent) variable, and the relevant anthropometric outcome as the dependent variable. Model 1 is adjusted for the baseline measure of the dependent anthropometric variable (for follow-up weight and BMI). Model 2 is as per model 1, additionally adjusted for age and sex. Model 3 is as per model 2, additionally adjusted for ethnicity, diabetes status and diabetes medication usage.

## Data Availability

The data presented here are available on request from the corresponding author, but are not publicly available due to restrictions related to patient confidentiality policies at our institutions.

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
