# Peer review of "Higher Waist Hip Ratio Genetic Risk Score Is Associated with Reduced Weight Loss in Patients with Severe Obesity Completing a Meal Replacement Programme"

_jpm, 2022, doi:10.3390/jpm12111881_

Round 1
Reviewer 1 Report
Handley et al analyzed the genotype of a cohort of obese patients to investigate the frequency of common risk alleles for obesity and type 2 diabetes-related traits. They calculated Genetic Risk Scores (GRS) for 6 traits and found that the waist-hip ratio WHR-GRS was inversely associated with weight loss of obese adults completing a 24-week milk-based meal replacement programme.
Although preliminary, this study highlights an interesting association between WHR-GRS and weight loss that needs to be further evaluated in other cohorts (including bariatric surgery etc). The paper is well written and the methodology is sound.
Comments:
- Please highlight the statistically significant values in the tables to make them more readable.
- Please define all abbreviations (for example the CADD score)
Reviewer 2 Report
Dale Handley et al. wrote an interesting paper entitled "Higher waist-hip ratio genetic risk score is associated with reduced weight loss in patients with severe obesity completing a meal replacement programme." The clinical study was designed correctly using very scientific research methods. The paper has outstanding scientific value. After addressing one major comment, a few minor comments, in my opinion, might be of value to be published in the Journal of Personalized Medicine.
Major comment
To formulate your final conclusion, "Patients with severe and complicated obesity with a genetic predisposition to central fat accumulation had less weight loss in a 24-week milk-based meal replacement programme….." on Fig. 1C, D, E, please show statistical analysis among tertiles (tertile 1 vs. tertile 2; tertile 1 vs. tertile, 3, tertile 2 vs. tertile 3). The p value presented on the graph, generated by ANOVA, shows only the diversity among 1, 2, and 3 tertiles.
Minor comments
-Please inform readers precisely whether your patients involved in the study were previously treated or not with bariatric surgery.
In the methods, you use formulation: “…..retrospective cohort study of adults with severe obesity attending our hospital-based bariatric service, who completed our milk-based meal replacement programme”, whereas in the discussion, “……those people with severe obesity who have a genetic tendency to store fat centrally respond less well to intensive dietary restriction. To our knowledge, this observation has not been described previously in bariatric patients undergoing a meal replacement programme.”
Any information about bariatric patients in the abstract
The part of the introduction concerning bariatric interventions should be shortened if your patients were not treated with bariatric surgery.
-The introduction is too long when compared with a very short discussion. The part of the introduction concerning genetic modification and its obesity-related complications moves to the discussion.
-Please explain abbreviations:
DQC and QC Line 196
CADD Line 201
-Please explain more precisely “milk-based meal replacement programme”; three phases each of 8 weeks (total 24 weeks)
-The results are not easily described to the readers.
-Line 232 “Patients with diabetes had lower body weight….” - this is not statistically confirmed (p=0.056; statistically significant p is less than 0.05).
-Line 234-235 “Variations in lipid profiles are likely due to a higher prevalence of statin use in patients with diabetes” – please insert a piece of information about statin use in each group in table 1 (n (%))
-Please inform the readers about comorbidities such as hypertension, thyroid diseases, and other diseases prevalent in obese patients. Table 1 should be enriched with this information.
- The authors divided patients into diabetic and non-diabetic (tab.1) and then did not continue their analysis. Maybe it is worth informing the readers whether your conclusions also apply to patients divided into diabetic and non-diabetic.
Round 2
Reviewer 2 Report
Thank you for your corrections.